# WHY ARE MODERN GANS POOR DENSITY MODELS?

## ABSTRACT

Modern generative adversarial networks (GANs) generate extremely realistic images and are generally believed to capture the true data distribution. In this work, we evaluate modern GANS as density models and ask whether they can be used for tasks such as outlier detection and generative classification. We find that the performance of state-of-the-art GANs is very poor on these tasks and is often close to (or worse than) random. For instance, a modern GAN that generates remarkably realistic samples when trained on CIFAR10, consistently assigns *higher likelihood to flat images* than to images from the training set.

To try and understand the source of this poor performance, we show that the likelihood that a GAN assigns to an input image is dominated by the quality of the GAN reconstruction when only the latent variable is optimized. Surprisingly, GANs often fail to reconstruct images from the training set in this scenario, while they are highly effective at reconstructing images outside the distribution. Taken together, our results indicate that modern GANs do not truly learn the underlying distribution, despite the impressive quality of the generated samples.

## 1 INTRODUCTION

Modern Generative Adversarial Networks (GANs, Goodfellow et al., 2020) have achieved extremely impressive results in image generation and manipulation (e.g. Brock et al., 2018; Karras et al., 2020b; 2021; Sauer et al., 2022; Pan et al., 2023). For many datasets of high-resolution images, naive observers find it difficult to determine whether a given image is real or a fake image generated by a modern GAN. The high perceptual quality of the generated images can also be measured numerically using metrics such as FID scores and the progress over the last decade of the success of GANs using this metric has been impressive. This progress would seem to suggest that the density model learned by GANs $p_\theta(x)$ has been gradually approaching the true input distribution $p_{\text{data}}(x)$.

But as has been pointed out repeatedly in the past (e.g. (Theis et al., 2015)), high-quality samples do not guarantee that the generator has learned a distribution that matches the true distribtion. A trivial example is a generative model that randomly samples a point from the training data: such a model would give excellent FID scores but will give zero probability to any example outside of the training set.

In this paper we wish to measure the extent to which modern GANs capture the true density of the data. Traditionally, generative models have been evaluated using both sample quality and additional tasks. Consider, for example, the Helmholtz machine, one of the first deep generative models of images. Published approximately 30 years ago, the authors demonstrated the success of a model trained on handwritten digits by: (1) showing samples from the model, (2) evaluating the log likelihood on held out data, and (3) training separate models for different classes and classifying new examples based on the class that gives the highest likelihood (Hinton et al., 1995; Frey et al., 1995). The motivation for the last task is the fact that the Bayes-optimal classifier is one that returns the category which maximizes the conditional density (Duda et al., 1973). Thus, we would expect a classifier that uses $p_\theta(x)$ as the density to achieve near optimal accuracy if $p_\theta(x) \approx p_{\text{data}}(x)$. Conversely, subpar accuracy serves as clear evidence that $p_\theta(x) \neq p_{\text{data}}(x)$. Indeed, Hinton et al. showed that the same model that generated realistic samples on handwritten digits also gave classification accuracies that were higher than a state-of-the-art classifier trained on the same data. We wish to determine whether a similar behavior holds in the case of modern GANs.

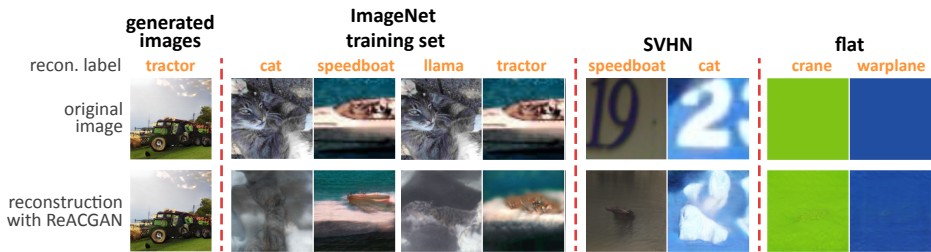

Figure 1: Modern GANs are typically bad at reconstructing training images when only the $z$ is optimized but can copy images completely outside the original training distribution. This is illustrated with ReACGAN (Kang et al., 2021) on ImageNet (with a reported FID of 15.65 Kang et al. 2022). The top row shows images generated by the GAN, from the training set, from SVHN (rescaled to $128 \times 128$) and images of a single color (flat). The second row shows ReACGAN's $z$-reconstruction of the images. Notice how the GAN can't reconstruct training images (optimization only $z$), whether conditioned on the correct label or on a different label, and the perceptual quality in both cases is equally bad. On the other, ReACGAN *can* reconstruct some images from SVHN and all flat images, even though they were never observed during training.

When we use these additional tasks to measure the success of modern GANs, we find that *the density models they learn are remarkably poor.* Their performance is worse than chance on outlier detection and worse than simple baselines on classification (e.g. $\approx 30\%$ accuracy on CIFAR10). This is despite the fact that the same GANs generate highly realistic samples for these datasets. In order to explain these failures, we show that the likelihood that a GAN assigns to a new image is dominated by what we call the "$z$-reconstruction error": the ability of a GAN to reconstruct an image when all parameters are fixed and we optimize over the latent variable $z$.

Take for instance the $z$-reconstruction results with ReACGAN (Kang et al., 2021) on ImageNet shown in Figure 1. The GAN is able to generate realistic samples, but is unable to reconstruct training images when only $z$ is optimized. Furthermore, the quality of the $z$-reconstruction when supplying the correct labels to the GAN or the wrong labels is equally bad, as shown in the left two columns of the training set for the correct labels and the right two columns for the wrong labels. Because the GAN frequently assigns similar likelihood to images in the correct class and the incorrect class, its performance on classification is very bad. Moreover, ReACGAN is able to $z$-reconstruct images made up of only a single color or from SVHN much better than images from the training set, even though they were never observed in training. This directly impedes the GAN in outlier detection, as it gives higher likelihood to flat images (outside the training distribution) than to test images. In our experiments, we observe that a range of state-of-the-art GANs on different datasets imitate these flaws, explaining why they perform so poorly on the tasks of classification and outlier detection.

The behavior described above defies what would usually be expected of good density models. If anything, images from the training set should be inside the support of the model while those from completely different distributions should be outside, which is apparently not the case for many modern GANs. All together, our results indicate that modern GANs in fact capture a fundamentally different distribution from that of the training data.

## 2 METHODS

GANs transform samples from a low dimensional latent space, $\mathcal{Z}$, into the (typically larger) dimension of the data $\mathcal{X}$, effectively describing a manifold. This corresponds to a generative model of the form $x = G_\theta(z)$ where $z$ is sampled from a base distribution $p(z)$ which, together with the mapping function $G_\theta$, determine the density of high dimensional samples $x \in \mathcal{X}$.

As defined, this distribution will give zero density to any sample not on the manifold. However, for many GANs almost all training points *are not* on the manifold, as shown in Figure 2. If almost all points of interest are not part of the manifold, then the performance of the GAN on any inference

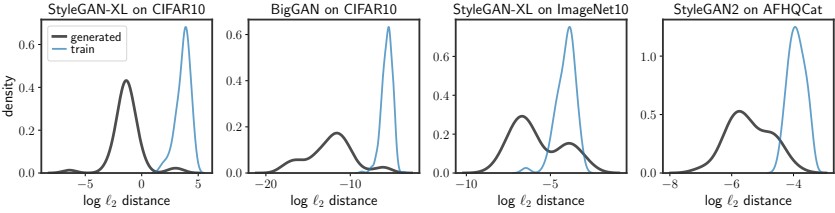

Figure 2: Euclidean distance of reconstructions of generated and train images under different GAN-dataset pairs (when only $z$ is optimized). In all examples, the reconstructions errors of training images are noticeably larger than the same for images generated by the GAN, implying that most training images are not inside the GAN manifold.

task will be essentially the same as random chance. It is therefore necessary to relax this definition of the density captured by GANs for successful inference.

Many authors augment the above density with an observation model: $x = G_\theta(z) + \eta$ where $\eta$ is observation noise and is usually assumed to be Gaussian with variance $\frac{1}{\gamma}$ (Wu et al., 2016). The probability of $x$ can now be rewritten as:

$$p_\gamma(x) = \int p(z) \, p_\gamma(x|G_\theta(z)) \, dz \tag{1}$$

where $p_\gamma(x|G_\theta(z))$ is the observation model. We will call this new probability the *relaxed likelihood* and will slightly generalize by assuming that the observation model takes the following form:

$$p_\gamma(x|G_\theta(z)) \propto \exp\left[-\gamma \cdot d(x, \, G_\theta(z))\right] \tag{2}$$

where $d(\cdot, \cdot)$ is a (possibly asymmetrical) distance function. In the case of a Gaussian observation model, the distance function is the $\ell_2$ norm.

Using such an observation model, the relaxed likelihood is consistent with that of the GAN at the limit $\gamma \to \infty$ but has full support as long as $\gamma$ is finite, unlike the distribution implied by the noiseless GAN. In our experiments, we will investigate the quality of the density models implied by GANs when $\gamma$ is finite.

## 2.1 ANNEALED IMPORTANCE SAMPLING

Calculating the relaxed log-likelihood involves solving the integral in Equation 1. Unfortunately, analytically solving this integral is intractable for the GANs that we study. Instead, approximate methods such as Markov chain Monte Carlo (MCMC) must be used in order to calculate the log-likelihood. In particular, Wu et al. showed that annealed importance sampling (AIS, Neal 2001) can be used to accurately approximate the log-likelihood of GANs. Broadly speaking, AIS is an MCMC approach that uses multiple intermediate distributions in order to estimate normalizing constants.

Let $f(z)$ be a target un-normalized distribution. An AIS chain is defined by an initial distribution $Q_0(z) = q_0(z)/Z_0$ whose normalization coefficient is known, together with $T$ intermediate distributions $Q_1(z), \cdots, Q_T(Z)$ such that $Q_T(z) = q_T(z)/Z_T = f(z)/Z_T$. Each step of the chain further requires an MCMC transition operator $\mathcal{T}_t$ which keeps $Q_t(z)$ invariant, such as the Mahalanobis-Adjusted Langevin algorithm (MALA) or Hamiltonian Monte-Carlo (HMC).

Beginning with a sample from the initial distribution $z_0 \sim Q_0(z)$ and setting $w_0 = 1$, AIS iteratively carries out the following steps:

$$w_t = w_{t-1} \cdot \frac{q_t(z_{t-1})}{q_{t-1}(z_{t-1})} \qquad z_t \sim \mathcal{T}_t(z|z_{t-1}) \tag{3}$$

The importance weights $w_T$ aggregated during the sampling procedure are an unbiased estimate of the ratio of normalizing coefficients, such that $\mathbb{E}[w_T] = Z_T/Z_0$.

Given an input image $x$, the relaxed likelihood can be calculated through AIS by setting $f(z) = p(z)p_\gamma(x|G_\theta(z))$ as the target distribution, such that $p_\gamma(x)$ is the corresponding normalization

constant $\mathcal{Z}_T$. We follow Wu et al. and use the following intermediate distributions:

$$q_t\left(z\right) = p\left(z\right) \cdot p_\gamma\left(x\mid G_\theta(z)\right)^{\beta_t} \tag{4}$$

where $\beta_t > \beta_{t-1}$, $\beta_0 = 0$ and $\beta_T = 1$.

In practice, we would like to estimate the likelihood in log-space to avoid numerical difficulties such as underflows, i.e. to calculate $\log p_\gamma\left(x\right)$. Calculating the log of the importance weights as described above is straightforward, however Grosse et al. have shown that doing so results in a stochastic lower bound of the log-likelihood. As the number of intermediate steps $T$ increases, this stochastic lower bound becomes tighter and converges to the true log-likelihood.

## 2.2 INFERENCE

We will use two tasks as a manner of testing whether GANs have learned the distribution underlying the training data: (1) generative classification and (2) outlier detection (OD).

**Generative Classification**  In generative classification, it is assumed that *different* parametric distributions were learned for each class $c \in \mathcal{C}$. The optimal classification estimator in this setting is to return the class with the highest conditional likelihood $p_\theta(x|c)$.

Some of the GANs that we investigate use parameter sharing in the models for different classes. For example, in StyleGAN-XL, a single GAN is learned for all classes, but the output of the GAN is conditioned on a one-hot vector that encodes the desired class. In such cases, we define the density that a model gives to a particular class as

$$p_\gamma\left(x|c\right) = \int p\left(z\right) p_\gamma\left(x|G_\theta\left(z,c\right)\right) dz \tag{5}$$

**Outlier Detection**  The simplest approach towards outlier detection assumes that outliers arise from some basic distribution (e.g. uniform), while inliers are generated from the learned distribution (Barnett, 1978; Barnett et al., 1994; Bishop, 1994; Zong et al., 2018). This setting corresponds to labeling any point with log-likelihood less than some predefined threshold $\tau$ as an outlier. This definition gives rise to the estimator $\hat{o}_\theta(x) = \mathbf{1}[p_\theta(x) < \tau]$ which returns 1 when $x$ is assumed to be an outlier. We will use the area under the ROC curve (AUC) to evaluate the performance of the different models on OD.

## 3 EXPERIMENTS

**AIS**  In all of the following experiments, we used AIS with an HMC transition kernel and 500 intermediate distributions and 8 chains. For more details and analysis of accuracy, see Appendix A.2.

**Relaxed Log-Likelihood**  For all experiments, a Gaussian observation model was used (see Appendix B.2 for other observation models). $\gamma$ was set as the inverse variance of the distance between training samples and the GAN's reconstruction of those images, corresponding to maximizing the relaxed log-likelihood with respect to the training images under a Gaussian distribution.

**Datasets**  The datasets considered are CIFAR10 Krizhevsky et al. (2009), AFHQ (Choi et al., 2020) and ImageNet (Russakovsky et al., 2015).

Because of the large computation cost of calculating the likelihood for each sample, we show results only for a small subset of the test data: 400 samples on CIFAR10/AFHQ, and 200 for ImageNet. Additionally, the results on ImageNet are for a subset of the dataset containing only 10 classes, which we call ImageNet10 (exact details in Appendix A). In the OD task, we use images of one color (which we call "flat"), SVHN and images from classes other than those being conditioned as the outliers.

**Models**  In our experiments, we consider the following pre-trained GANs: StyleGAN-XL (Sauer et al., 2022), BigGAN-DiffAug (Zhao et al., 2020), StyleGAN2-ADA (Karras et al., 2020a),

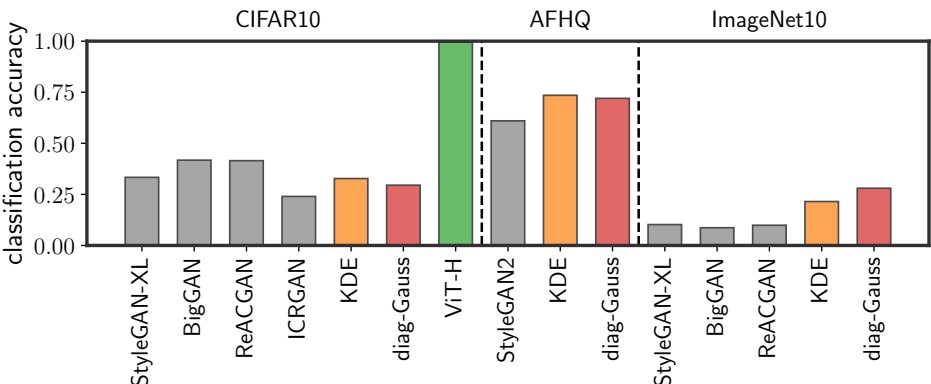

Figure 3: Generative classification accuracy of the different GANs (in gray) compared to our simple baselines (orange and red). All of the GANs are on par, or worse than, generative classification using simple baselines. Additionally, the classification accuracy of ViT-H (Dosovitskiy et al., 2020) on CIFAR10 (in green) shows how far the GANs are from current state-of-the-art classifiers.

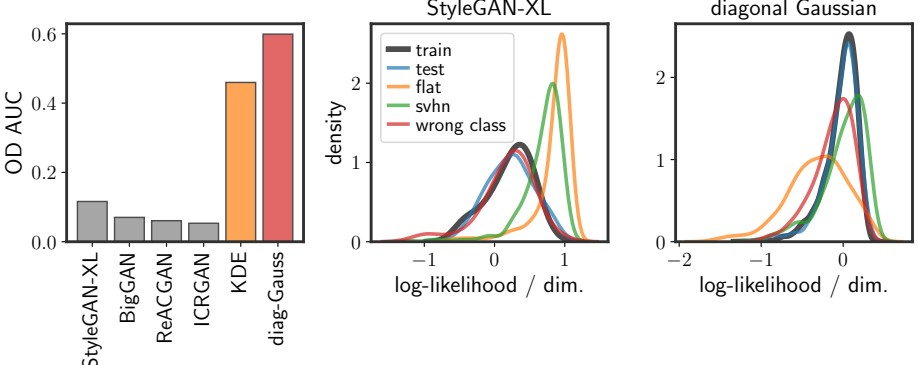

Figure 4: **Left**: Outlier detection performance on CIFAR10 (truck class) of the different GANs, compared with the baselines. **Right**: histograms of the log-likelihoods assigned by StyleGAN-XL (center) and a diagonal Gaussian (right) to different partitions of the data. Note that StyleGAN-XL gives flat (orange), SVHN (green) and images from the wrong class (automobile, red) higher likelihood than even the training images (black). On the other hand, the diagonal Gaussian is at least able to differentiate between flat images and test images.

BigGAN-ICR (Zhao et al., 2021), ReACGAN (Kang et al., 2021) (the last 2 we use the implementations available from Kang et al. 2022). The results for CIFAR10 and ImageNet10 use class-conditional GANs, while those on AFHQ use GANs trained separately on each class. These GANs were chosen as they all perform extremely well under the standard evaluation protocols - for instance, StyleGAN-XL has an FID of 1.52 on ImageNet and an FID of 3.35 on CIFAR10.

**Baselines** We will compare the GANs to two simple baselines: (1) a diagonal Gaussian and (2) a kernel density estimator (KDE) with an isotropic Gaussian kernel and variance around 0.05. Both of the baselines are fitted to the training data (or a subset of it), carrying out inference in exactly the same manner as the GANs.

## 3.1 RESULTS

Figure 3 shows the classification accuracy of various GANs on different datasets. Notably, the classification accuracy of the GANs is very low, typically lower or on par with our simple baselines.

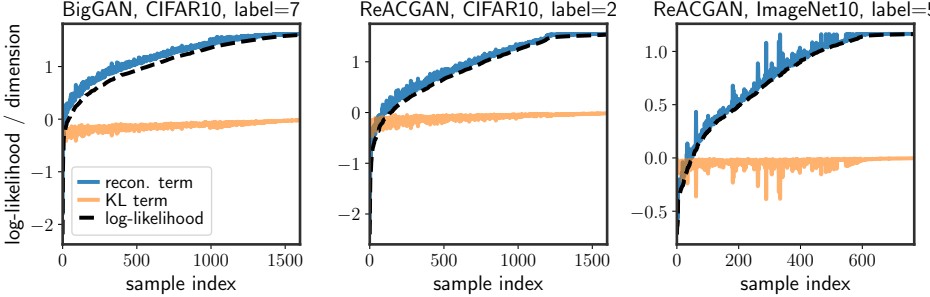

Figure 5: The log-likelihood of samples plotted in ascending order for different dataset-GAN pairs. The full log-likelihood (dashed line) is the sum of two terms: a reconstruction term (in blue) and a KL term (in orange). In all of the cases we investigated, the KL term remains more or less constant, while the reconstruction term varies between samples. These results imply that the log-likelihood is mostly dependent on the reconstruction quality of the samples.

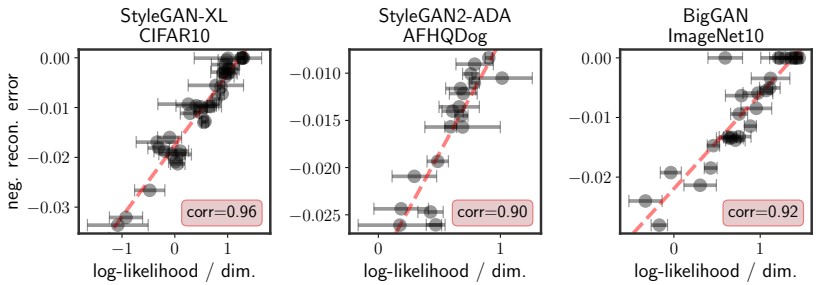

Figure 6: The negative $z$-reconstruction error scattered against the log-likelihoods (divided by the dimension of the data) found using AIS for various dataset-GAN pairs. The errorbars represent $\pm 1$ standard deviation of the importance weights found using AIS. The dashed red line in the linear fit of the $z$-reconstruction error to the likelihood. The reconstruction error is highly correlated with the log-likelihood, typically around 0.9, hinting that it serves as a good stand-in for the true relaxed log-likelihood. Correlations for all pairs of data sets and GANs available in Appendix B.3.

Figure 4 (left) shows the results of OD on CIFAR10. The performance of the GANs on these tasks is very low - much lower even than randomly selecting outliers. Figure 4 (center) shows why this is the case: the likelihood assigned to flat and SVHN images is higher than to the test and train images. This means that any threshold chosen will assign more test images as outliers than the actual outliers. Worse, most of the *training images* will also be considered outliers!

The results shown in Figure 4 are only conditional on a single class - the CIFAR10 "trucks" class; more results can be seen in Appendix B. Furthermore, the performance of the GANs on ImageNet10, AFHQCat and AFHQDog also fall below the baselines and is available in Appendix B.

## 4 ANALYSIS

The results above indicate that modern GANs are not effective density estimators. But why is this the case?

The relaxed log-likelihood can be rewritten as follows:

$$\log p_\gamma(x) = \mathbb{E}_{z|x}\left[\log p_\gamma\left(x|\,G_\theta(z)\right)\right] - D_{\text{KL}}\left(p_\gamma\left(z|x\right)\,||\,p(z)\right) \tag{6}$$

The first term is related to how well the model can reconstruct the given image and is easier to analyze while the second term is the divergence between the model's prior and posterior.

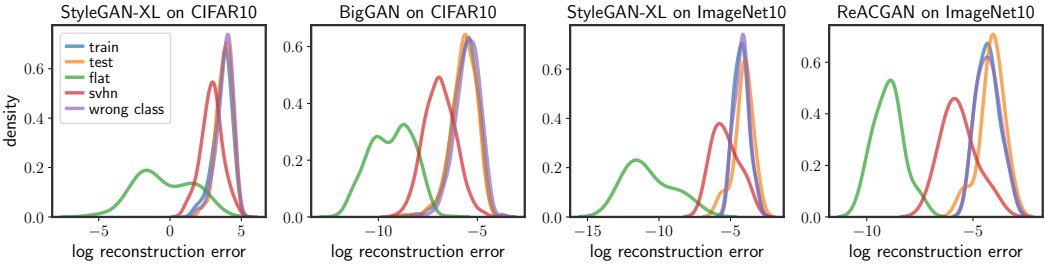

Figure 7: Histograms of the log $z$-reconstruction error (lower is better) of different GANs-dataset pairs on different partitions of the data. Notice how flat and SVHN images consistently achieve lower reconstruction error, even lower than training images, explaining why GANs struggle with outlier detection. Moreover, the reconstruction of training and test images are almost identical to the reconstruction error when the GAN is provided with the wrong label, leading to poor accuracy in generative classification.

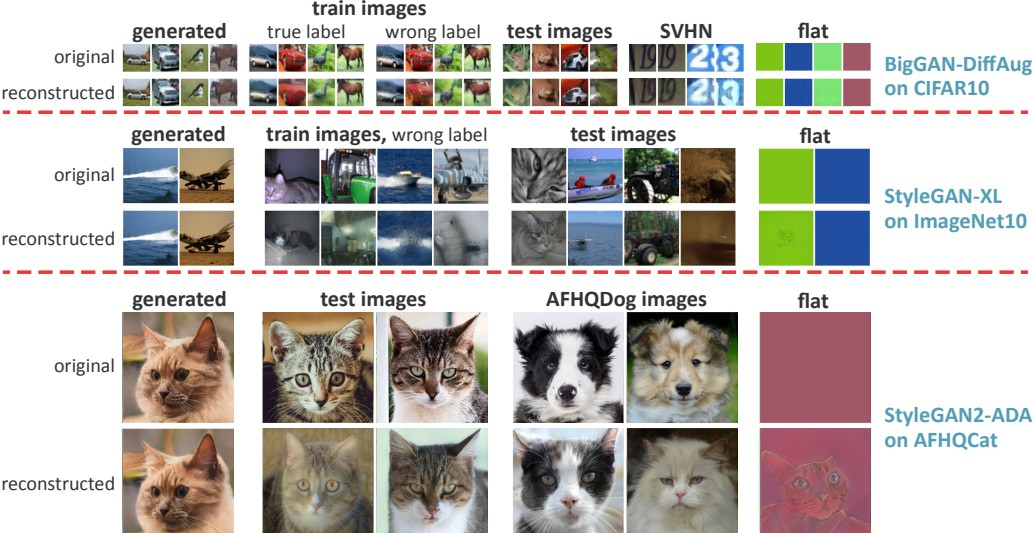

Figure 8: Best reconstructions for various dataset-GAN pairs. The top rows are results for BigGAN-DiffAug trained on CIFAR10 (FID 3.35, zoom in to see details), the middle rows are StyleGAN-XL on ImageNet10 (FID 1.52), and the bottom rows are StyleGAN2-ADA on AFHQCat (FID 3.55). For each GAN, the top row consists of original images and the bottom is the GAN's reconstruction. All of these GANs are able to $z$-reconstruct flat images remarkably well, even though such images weren't seen during training, which is the reason they under-perform on OD. On the other hand, the reconstruction of training and testing images is quite poor with all GANs. Moreover, in most cases the reconstruction of training images is equally bad whether the labels supplied to the GAN are the true labels or incorrect labels, explaining why they struggle in the task of generative classification.

We can look at the relative importance of these terms on the log-likelihood in order to gain a better understanding of the behavior empirically observed in the previous section. In Figure 5, the reconstruction and KL terms are plotted next to the log-likelihood. Notice how the difference in log-likelihood between different samples is mostly determined by the reconstruction term, while the KL term is almost constant.

### 4.1 ANALYSIS THROUGH z-RECONSTRUCTION QUALITY

As shown above, the relaxed log-likelihood is largely dependent on the reconstruction term from equation 6. However, it is computationally costly to calculate this error and doesn't enable further interpretation of the failure modes. Instead, we can look at the mode of the posterior, centered around the GAN's best reconstruction of the input image $x$.

Let $\hat{z}$ be the latent code corresponding with an output that is closest to the input image:

$$\hat{z} = \arg\min_z d\left(x,\ G_\theta(z)\right) \tag{7}$$

We will call $-d\left(x,\ G_\theta(\hat{z})\right)$ the *negative z-reconstruction error* of the GAN[1]. We emphasize that this reconstruction error is the error when all parameters of the GAN are held fixed, and we only optimize over the latent code $z$. This is in contrast to many modern GAN inversion methods (e.g. (Sauer et al., 2022)) in which different parameters of the model may be changed during the inversion to better fit the input.

In practice, this $z$-reconstruction error is highly correlated with the relaxed log-likelihood in the settings we have examined, as shown in Figure 6. As such, analyzing the quality with which a GAN can copy images is a good proxy for investigating how likelihood is assigned by the GAN to the images.

### 4.2 z-RECONSTRUCTION PERSPECTIVE OF GAN FAILURE

Figures 8 and 7 show the $z$-reconstruction error of different GANs visually and quantitatively, respectively. These results hint at the reasons behind the poor performance of GANs on both classification and OD.

**Failure on Classification**  The process of generative classification involves calculating the likelihood of the image under all possible classes, returning the class whose likelihood is maximal. However, as can be seen in Figure 7, the $z$-reconstruction errors of all of the GANs when supplied with the wrong label (in purple) or with the correct label (in blue and orange) are all concentrated in the same area. In other words, the likelihood given by the GAN to a test image is approximately the same for all classes, hurting the classification accuracy of the GAN.

**Failure on OD**  Figures 8 and 7 both illustrate how the quality of the $z$-reconstructions of flat and SVHN images is overwhelmingly better than even that of the training images. This can be understood from a geometric stand point: the GAN manifold passes closer to flat images than to images from the train set. Therefore, the intuitive definition of outliers as points far from the GAN manifold estimates most training/test points as outliers.

## 5 RELATED WORKS

This work follows a line of works (e.g. Nalisnick et al. 2018; Fetaya et al. 2019; Kirichenko et al. 2020) that show that many modern generative models do not truly capture the underlying data distribution. As far as we know, GANs have not been analyzed in this manner, despite their ability to generate images perceptually similar to natural images, as there is no direct access to the model likelihood.

There is a vast literature regarding inference using GANs and evaluating their performance (e.g. Heusel et al. 2017; Sajjadi et al. 2018; Ravuri & Vinyals 2019; Webster et al. 2019; Naeem et al. 2020; Borji 2022; Ravuri et al. 2023). In this space of works, ours is most similar to that of Ravuri & Vinyals 2019 in the sense that both methods use performance on classification as a means of ascertaining whether GANs have learned the correct distribution or not. A key difference between the approaches is that Ravuri & Vinyals 2019 suggested training a *separate, discriminative classifier* on data generated by the GAN, whereas we utilize purely *generative* classification and OD as a means to understand whether the GANs have learned the correct distribution.

---

[1]See Appendix A.3 for more information on the GAN inversion method

Utilizing the fact that GANs are generative models for classification, regression and OD performance is not new (e.g. Schlegl et al. 2017; Donahue & Simonyan 2019; Kang et al. 2021; Nitzan et al. 2022). However, all such methods consider use an additional component such as an additional encoder which is not part of the generative process of the GAN. In contrast, we argue that if GANs accurately capture the data distribution, their generator *alone* should be enough to achieve near-optimal performance.

## 6    LIMITATIONS

The main limitation of this work is that calculating the AIS log-likelihood and reconstructing images using GANs are both computationally expensive tasks. As such, it is hard to evaluate large quantities of data. Fortunately, the sample sizes do not necessarily need to be huge in order to expose existing problems in models, as shown in this work.

Furthermore, while bad performance on inference tasks is evidence that a model doesn't represent the true distribution, it does not directly translate into the distance between said distributions. That is, increasing the accuracy of generative classification (for instance) doesn't guarantee better density estimation. Concretely, while our baselines had better accuracy they were definitely worse generative models than the GANs in most other aspects. However, we should still expect that generative models better at classification *while still* performing well under other evaluations will be better generative models than those that don't.

Finally, our (and previous works') definition of the likelihood adds a degree of freedom not originally present in the trained generative model in the form of the observation model. The behavior of GANs with different observation models could vary wildly and will influence performance on tasks outside of image generation. However, we believe that using the simple Gaussian observation is sufficient for most purposes. Specifically, we should expect GANs to reconstruct training images well and images from different distributions badly, which *is* captured by the Gaussian observation model. Moreover, if the relaxation of the likelihood isn't used, GANs would not be able to perform well in any task, as argued before.

## 7    DISCUSSION

An ideal density estimator maximizes the likelihood of the training distribution while minimizing the likelihood at every other portion of space. This kind of ideal behavior should result in low density for distributions completely different from the training distribution and high likelihood on held-out data (as long as they are from the same distribution as the training data). Said differently, in order for GANs to perform as good density models, we should hope that they will be able to give high likelihood to images from the training and test datasets, but not flat images or images of digits. Moreover, a GAN trained on one class of CIFAR10 or ImageNet should not give high likelihood to images in another class.

As we have shown, modern GANs on a range of datasets *do not* allocate their likelihood in this ideal manner. Leveraging the fact that the z-reconstruction ability of the GAN captures most of the variance in the log-likelihood of the GAN enables us to better understand *why* GANs fail. In particular, it seems that the GAN manifold passes far from the training distribution, contrary to what is typically believed. This hints that during training GANs optimize a different objective than ensuring that the manifold passes close to training points and warrants further exploration. In summary our results suggest that modern GANs learn a distribution that is very different from the distribution of the data.

## 8    REPRODUCIBILITY STATEMENT

To ensure the reproducibility of our results, all information on how the log-likelihood was calculated, how $z$-reconstructions were found and which data we used is available in the appendix. Furthermore, the AIS code we used will be made available in the Supplementary Material and, later on, publicly.

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

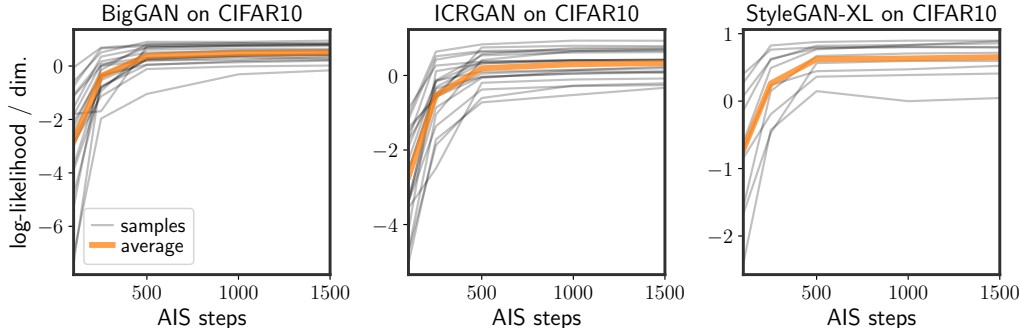

Figure 9: The estimated log-likelihood as a function of AIS steps, for different GANs on CIFAR10 on random test samples. The log-likelihood of most samples converges after 500 steps.

Zhengli Zhao, Sameer Singh, Honglak Lee, Zizhao Zhang, Augustus Odena, and Han Zhang. Improved consistency regularization for gans. In *Proceedings of the AAAI conference on artificial intelligence*, volume 35, pp. 11033–11041, 2021.

Bo Zong, Qi Song, Martin Renqiang Min, Wei Cheng, Cristian Lumezanu, Daeki Cho, and Haifeng Chen. Deep autoencoding gaussian mixture model for unsupervised anomaly detection. In *International conference on learning representations*, 2018.

## A  Implementation Details

### A.1  ImageNet10

Because of the computational cost of calculating the log-likelihood for each image, we used a small subset of ImageNet. In our experiments, we used 10 classes, which is why we called the subset ImageNet10. These classes are: warplane, sports car, heron, tabby cat, llama, vending machine, bullfrog, coffee mug, speedboat, and tractor.

### A.2  AIS Details

We follow the implementation of AIS from Wu et al. (publicly available in GitHub), reimplemented in PyTorch. When possible, we used the same settings as Wu et al.:

- The transition operator we used was HMC with 10 leapfrog steps and a Metropolis-Hastings (MH) adjustment. During sampling, the learning rate is initialized to $5 \cdot 10^{-2}$ and adjusted according to a moving average of the MH rejection rate

- During sampling, the intermediate distributions we used were:

$$Q_t(z) \propto p(z) \cdot p_\gamma \left(x \mid G_\theta(z)\right)^{\beta_t} \tag{8}$$

  $\beta_t$ was annealed according to a sigmoidal schedule

- In all experiments we use 8 chains in order to calculate the importance weights

**Choice of Number of Steps**  The bound on the log-likelihood approximated by AIS becomes tight and accurate only as the number of intermediate distribution and number of chains grows, respectively. However, AIS with many chains and intermediate distributions is incredibly computationally costly. Due to these considerations we use a relatively small number of intermediate distributions, while still ensuring accurate enough results.

Each chain used 500 intermediate steps. This number is in stark contrast to the 10,000 iterations used by Wu et al.. We chose this number by running multiple AIS chains with a differing number

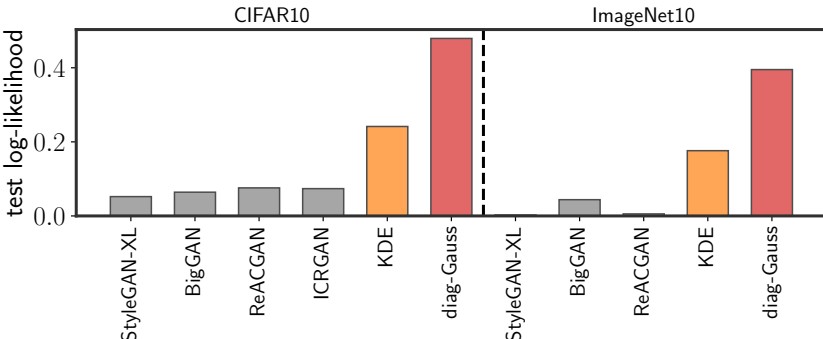

Figure 10: OD results when averaging over all labels. Here, again, the baseline models outperform the GANs investigated.

of intermediate steps, plotting the estimated log-likelihood as a function of AIS steps, as shown in Figure 9. The estimated log-likelihood typically asymptotes very close to the value reached after 500 iterations. The difference between the converged value and the one after 500 steps is much smaller than the resolution of log-likelihoods we are looking at, so this is a compromise between accuracy and computational cost.

Further justification for this is due to the comparison between AIS and GAN inversion in terms of gradient steps. Because of the leapfrog steps, a single iteration of AIS is similar to 10 gradient steps in GAN inversion. In all of our experiments, $\sim$1500 iterations were enough to converge during inversion, well below the 5000 gradient steps used during the AIS procedure.

### A.3 Reconstruction through GAN Inversion

There is a vast literature on the best way to reconstruct test images using GANs, also called *GAN inversion*. In this work we used a simple, albeit rather costly, approach in order to find the best possible reconstruction.

We used an optimization approach towards GAN inversion, using ADAM as the optimizer and a cosine schedule (similar to the scheme used by Sauer et al. in their implementation). To find better reconstructions, we sampled $\sim 1000$ images from the GAN and initialized the optimizer from the latent code of the image closest to the input image in $\ell_2$ distance. Furthermore, this process was repeated $\sim 8$ times for each image. Using this GAN inversion scheme, we were always able to invert images generated by the GAN (and frequently flat images as well).

Finally, note that for all experiments with the StyleGAN variants, the inversion took place in $\mathcal{Z}$ space, as the generative model is defined in terms of this latent space and not the $\mathcal{W}/\mathcal{W}+$ spaces.

### B More Results

### B.1 Outlier Detection Results

OD on ImageNet10 and AFHQ can be seen in Figure 11. As mentioned in the main text, for both of these datasets GANs underperform, similarly to CIFAR10.

The results shown in the main text and above on conditional GANs showed OD performance when the GAN is conditioned on a single label. Instead, we can think of the GAN as a mixture model over all labels, in which case it could be the case that OD on a single class didn't work simple due conditioning. The results in Figure 10 show OD performance when viewing the conditional GANs as mixture models. Here, again, the AUC of the GANs is very poor, even next to the simple baselines.

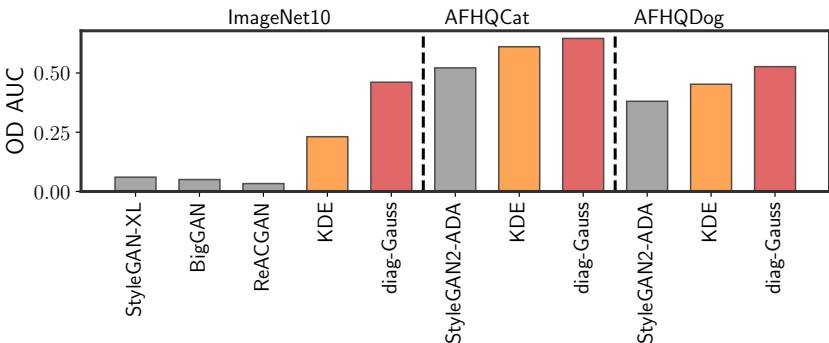

Figure 11: The GANs' performance on OD in ImageNet10, AFHQCat and AFHQDog, compared to the performance of the baselines.

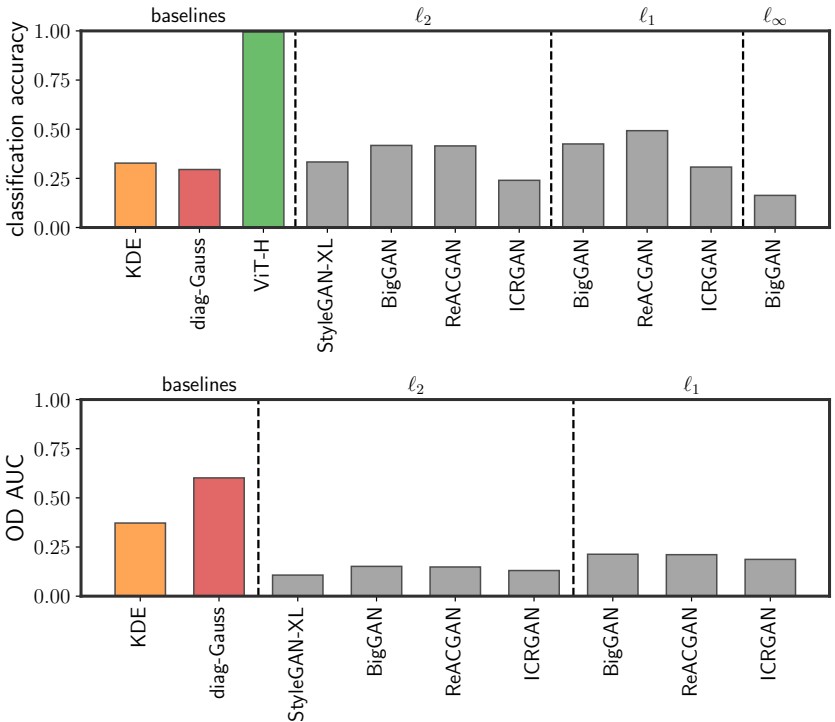

Figure 12: Classification (top) and OD (bottom) performance for various different distance functions. The GANs perform poorly with all distance functions explored.

|  | BigGAN | ReACGAN | ICRGAN | StyleGAN-XL | StyleGAN2-ADA |
|---|---|---|---|---|---|
| CIFAR10 | 0.93 | 0.86 | 0.89 | 0.95 | - |
| ImageNet10 | 0.92 | 0.98 | - | 0.9 | - |
| AFHQCat | - | - | - | - | 0.77 |
| AFHQDog | - | - | - | - | 0.9 |

## B.2 DIFFERENT DISTANCE FUNCTIONS

The relaxed likelihood as was defined in Section 2 is dependent on the particular distance function $d(x, G_\theta(z))$. In the body of the paper we only considered the Euclidean distance, however performance could in theory vary greatly if the distance function is changed. Figure 12 explores the use of the $\ell_1$ and $\ell_\infty$ norms. While the numbers vary slightly, the overall performance of GANs is still worse than the simple baselines.

## B.3 CORRELATIONS WITH RECONSTRUCTION ERROR

In almost all settings we explored, the negative reconstruction error is highly correlated with the AIS, as seen in the following table:

During calculation of the correlation between the negative reconstruction error and AIS, the images with the top 5% variance in importance weights were excluded. Additionally, the images considered for this calculation were train images, test images, images from SVHN and flat images.

