# OpenReview forum: "Why are Modern GANs Poor Density Models?"
_ICLR.cc/2024/Conference — Submitted to ICLR 2024_

### Official Review · Reviewer_SxfR · 2023-10-27

**Soundness:** 2 fair
**Presentation:** 2 fair
**Contribution:** 2 fair
**Rating:** 3
**Confidence:** 4

**Summary:**

The paper aims to elucidate why GANs perform poorly as density models for outlier detection and generative classification tasks with recent GAN models like BigGAN and StyleGAN. The authors use latent optimization as a tool in the investigation. They observed that GANs tend to assign higher likelihood to out-of-distribution images (e.g., flat images) compared to images from the training set. Additionally, the authors found that the quality of image reconstruction has a significant dominant over the KL divergence in likelihood assignment to input images. The experiments were limited to subsets of CIFAR-10, SVHN, and ImageNet due to the computational expense of the proposed method.

Overall, the paper is easy to read. However, there is a need for additional details, as indicated in the comments below. While likelihood assignment represents the primary valid contribution of the paper, there are still gaps that require clarification and more supporting details to bolster the claims made in the paper. Based on my current understanding, I would recommend rejecting the paper in this state.

**Strengths:**

The conclusion that GANs fail to learn the underlying distribution of training data is not new. Nevertheless, the paper does introduce some seemingly novel insights, particularly regarding likelihood assignment to input images.

**Weaknesses:**

However, the limitation to only a subset of the datasets raises concerns about the representativeness of their findings in the broader distribution. Furthermore, there is a missing piece of values of optimized z that the authors may have overlooked, which requires verification (as outlined in the questions below). Without addressing these issues, the validity of the contribution is in question.



1. The explanation of the z-reconstruction error is not entirely clear to me. Based on my intuition, it seems that the latent is optimized to maximize the likelihood of reconstructing the image, which is equivalent to minimizing the sample's reconstruction error. If my understanding is correct, there appear to be some concerns related to experimental design and the claims made when using the z-reconstruction method as a proxy for validating the model's density, particularly when working with limited data.  For more details. In Figure 1, the authors demonstrate that the GAN can reconstruct the generated images better than the training images. The results are predictable because the GAN was optimized during training to generate these specific images, making them more amenable to reconstruction and resulting in lower error. To explain further, if we consider the bad GAN model which collapses a specific mode of data distribution only, this GAN excels at reconstructing these samples but may struggle with reconstructing images outside of this distribution mode. However, when the authors use only a limited dataset, it's possible that the examples representing the mode the GAN learned are not included.  My point is that a comprehensive investigation requires access to the full dataset to ensure robust support for the findings. Could the authors provide a plot of the full dataset? This request also relates to Figure 2, as there might be training examples that the GAN could collapse into. If these examples are not utilized, the complete picture of the training distribution may not be accurately represented.


2. The reconstructions of SVHN and flat images in Figure 1 are equally poor when compared to the training examples to me. It is clear that GANs manage to capture the global color appearance and shape, but they struggle with preserving the finer details of the images. Even in the case of flat images where we can still discern patterns (indicating they are not entirely flat) from the training distributions. Only the reconstructions of the generated images from the model appear to be well, largely consistent to the reasons I mentioned earlier. Therefore, it might not be sufficient to assert that the model "can copy images completely outside the original training distribution," as it appears that the reconstruction behavior remains consistent across most images, and this claim should be approached with caution.


3. One crucial aspect that I couldn't find in the author's discussion in the paper is the value of the optimized z. GANs are optimized with a specific latent distribution, such as a Gaussian within a defined range, and they may behave differently when confronted with latent variables that fall outside of this distribution. It would be beneficial if the author could provide more information about the latent distribution used for generating images in the investigated GAN models. Also, comparing the optimized z values for SVHN and flat images to those optimized for the generated samples and training images?. This comparison is only valid when the optimized z values fall within the latent distribution that the GAN was originally trained with. The choice of optimized z values could significantly impact all other results in the paper. If it turns out that the optimized z values are far outside the actual latent distribution required, this could raise questions about the validity of the findings. A similar concern could be applied to Figure 5. If the optimized z values are indeed found to be outside the latent distribution, it may result in KL divergence values that remain relatively constant, dominated by the reconstruction term, which tends to be better optimized.

**Questions:**

**More comment and questions**

* In Figure 1, could the authors include the results of other GAN models on the same images to strengthen their claim?

* Could the authors clarify the distinction between density and likelihood as defined in the paper? Furthermore, it's not clear why the density is greater than 1 in Figure 4?

* The rationale behind dividing likelihood by the dimension in many plots in the paper could also be elaborated upon?

* Can authors explain how to sample and generate of “flat images” in Figure 4, Figure 7?

---

> ### Author Response · Authors · 2023-11-14
> **Response to Reviewer SxfR**
>
> 1. We disagree with your interpretation of figure 1. You write that “The results are predictable because the GAN was optimized during training to generate these specific images,” but the GANs were optimized to generate TRAINING images, not GENERATED images, and figure 1 shows that training  images are not reconstructed well. Regarding the hypothesis that the GANS might be reconstructing well a subset of the training images, please see figure 2 which shows the histogram of the reconstruction error. You can see that for StyleGAN-XL on CIFAR10 almost all training images have an L2 reconstruction error that is much larger than that of the reconstruction error for the generated images (note the log scale on the x-axis).
> 2. We agree with the reviewer that such claims should be approached with caution. However, in this statement was made due to the results shown in figure 7, in which flat and SVHN images clearly have (in general) a much lower reconstruction error (please note that the errors are in log scale!). Of course, we do not want to mislead readers and if the reviewer still feels that this is inappropriate, we will modify the statement in the introduction.
> 3. We agree that the norm of the latent code influences the likelihood. Note however, that  the first layer of many StyleGANs (such as StyleGAN-XL and StyleGAN2 which we used) is an InstanceNormalization layer which normalizes the latent code to have a norm of 1 and our z-reconstruction also satisfies the same constraint. In other words, for these StyleGAN models all the latent codes that are found for out of distribution images are of the same norm as the ones within distribution. For the other models, we did not see a significant difference between the norms of latent codes for in-distribution vs out-of-distribution. We agree that these results should be included in a future version.
>
> ---
>
> Answering the reviewers further questions:
> - In this research we use density and likelihood interchangeably (as is common in this type of research), always to mean the density function $p_\theta(x)$. The densities of continuous random variables can be larger than 1.
> - As is common in the field, the likelihood is divided by the dimension to give an indication of information content per pixel.
> - As written in section 3, flat images are just images of a single color. To generate these images, we sample 3 values from a uniform distribution in the range (0, 1), all pixels in the red channel of the image are equal to the first value, pixels in the green to the second and the blue channel is equal to the last. Examples of these images can be seen in figures 1 and 8, always at the right-most side of the figure.

---

### Official Review · Reviewer_kwJr · 2023-10-31

**Soundness:** 2 fair
**Presentation:** 2 fair
**Contribution:** 2 fair
**Rating:** 5
**Confidence:** 4

**Summary:**

The authors present a methodology to answer the question of how well modern GANs estimate the density of the ground truth distribution. Since GANs do not have a tractable density, the authors propose a relaxed likelihood framework by adding noise to the push forward of the generator with a latent vector defining the density everywhere on the manifold. Predicated on this augmentation, the likelihood of this density can be solved numerically with annealed importance sampling. Furthermore, the authors empirically test the quality of density estimation given by the Generator by evaluating its performance on two tasks, outlier detection and generative classification, as well as provide analysis to understand why GANS is not performing well on these tasks.

**Strengths:**

1). The authors provided sufficient empirical evidence that modern GANs perform poorly on outlier detection and generative classification tasks.

2). The plots are well done and easy to read.

3). Parts of the paper are well written.

**Weaknesses:**

1). Some of the assumptions and motivational arguments could benefit from more explanation.
* The sentiment of GANs should be good density estimators because they are capable of generating a realistic sample. A justification is required because GANs are not trained with a well-defined density, so is it reasonable to assume they can be good at estimating something they were not trained to do?
* "The GAN is able to generate realistic samples but is unable to reconstruct
training images when only z is optimized." There should be some justification here explaining why this is possible even though the optimization problem defined in eq. (7) is np-hard [Inverting Deep Generative Models].
One layer at a time](https://proceedings.neurips.cc/paper_files/paper/2019/file/24389bfe4fe2eba8bf9aa9203a44cdad-Paper.pdf).

2). Analysis of the z-reconstruction error term
* This error will exist because a k-dimensional vector cannot represent all of $R^ {n}$, this has been extensively studied in the field of GANprior [Compressed Sensing using Generative Models](http://proceedings.mlr.press/v70/bora17a/bora17a.pdf). The density truly lies only in the range of $G(z)$, but since noise was added to the manifold to have a well-defined density over $R^{n}$, it is unclear if the poor recon error and likelihood are dominated by the noise added to the manifold or the n-k dimensions of the manifold z cannot represent. A noise-to-signal ratio metric would mitigate this issue.

3). Problem formulation of the proposed method
* AIS-logliklihood calculation is unprincipled because GANS do not have tractable density, so noise must be added to the system to define a density.

4). At times, the paper read as if it were attempting to answer too many questions.
* The papers consistently switch between the question of how well Gans estimate density and how poor they estimate density. The paper would be easier to follow if it were read as first demonstrating evidence for question 1, then moving onto the next question.

**Questions:**

1). Could the authors provide metrics for noise to signal ratios of the observed model $x=G_{\theta}(z) + \eta $ for training and test sets?

2). Could the authors please justify why this is a reasonable assumption? "The GAN is able to generate realistic samples but is unable to reconstruct training images when only z is optimized," even though the defined optimization problem is np-hard.

3). Could the authors provide a justification on why we should expect GANs to be good at estimating density when they are not trained specifically for such a task?

4). Could the authors please describe the procedure for how the flat images are created? As well, what is the definition of flat? Is this a reference to the manifold of the image?

---

> ### Author Response · Authors · 2023-11-14
> **Response to Reviewer kwJr**
>
> We thank the reviewer for their comments and suggestions.
>
> As we understand, the major concerns of the reviewer are with how well GANs capture densities and the definition of the z-reconstruction. For the first of these, please see the general comment we made to all reviewers. Briefly, it has previously been shown theoretically that the optimum of a GAN is the true training distribution.
>
> As the reviewer mentioned, fully solving the z-reconstruction optimization problem is NP-hard. However, we use methods common to many other studies, in particular gradient descent with multiple random restarts. As is evident from figure 1 and 8, this method DOES give almost perfect reconstructions for images generated by the GAN. This means that despite the worst-case hardness results, it is possible to approximately solve the z-reconstruction problem for many cases. On the other hand, using exactly the same method, training/test reconstructions are visibly very different from the original images. Figure 2 demonstrates this quantitatively - the errors of the optimization process are much larger for training images than for images generated by the GAN itself. Please let us know if this does not resolve your issues with the z-reconstruction.
>
> ---
>
> Answering the reviewer’s questions:
> 1) The values for the noise that we used and their SNRs can be seen in the following table, which will be added to the appendix:
>
> | GAN |  Dataset |  Value of $\\gamma^{-1}$ | SNR |
> | ---- | ------------ | ----------------------- | --- |
> | BigGAN |  CIFAR10 |  0.0062 | 30.5 |
> | ICRGAN | CIFAR10 | 0.0076 | 24.9 |
> | ReACGAN | CIFAR10 | 0.0071 | 26.4 |
> | StyleGAN-XL | CIFAR10 | 0.015 | 14 |
> | BigGAN | ImageNet10 |  0.012 | 12.3 |
> | ReACGAN | ImageNet10 | 0.015 | 9.6 |
> | StyleGAN-XL | ImageNet10 | 0.016 | 9.1 |
> | StyleGAN2-ADA | AFHQCat | 0.022 | 10.2 |
> | StyleGAN2-ADA | AFHQDog | 0.021 | 8.7 |
>
> 2) and 3) Please see our answer regarding the z-reconstruction at the top of this comment.
> 4) As written in section 3, flat images are just images of a single color. To generate these images, we sample 3 values from a uniform distribution in the range (0, 1), all pixels in the red channel of the image are equal to the first value, pixels in the green to the second and the blue channel is equal to the last. Examples of these images can be seen in figures 1 and 8, always at the right-most side of the figure.

---

> > ### Comment · Reviewer_kwJr · 2023-11-21
> > **Official Comment**
> >
> > I would to thank the authors for their response.
> >
> > 1). Is the reported SNR in terms of dB's?
> >
> > 2). Thank you, for addressing solving an np-hard optimization problem when reconstructing the latent space.
> >
> > Based on the author's rebuttal I will change my score from a 3 to a 5.

---

> > > ### Author Response · Authors · 2023-11-23
> > >
> > > Thank you for the updated score.
> > > The SNRs are **not** in terms of dBs, sorry for the confusion.

---

### Official Review · Reviewer_PGsS · 2023-11-06

**Soundness:** 2 fair
**Presentation:** 3 good
**Contribution:** 2 fair
**Rating:** 5
**Confidence:** 4

**Summary:**

This work investigates GANs as a density estimator and reveals that state-of-the-art GANs are very poor on density-related tasks. Hence, GANs do not truly learn the underlying distribution. To understand this phenomenon, the authors show that if using model inversion, the likelihood is dominated by the so-called "z-reconstruction error", which GANs often perform better for out-of-distribution data than the training data.

**Strengths:**

The problem being investigated is interesting and potentially important for practitioners. The writing is clear and most of the arguments are supported by experiments.

The authors investigated GANs in both classification and OOD detection tasks. The link between the performance and the z-reconstruction error is interesting and novel to me.

**Weaknesses:**

## The motivation is relatively weak:
As the authors stated in Section 5, this phenomenon is already known for various generative models. Hence, it's not unexpected that GANs are also not good. I am not entirely convinced that this is important for GANs nowadays as nobody actually uses them for inference tasks.

On the theoretical side, GANs as density estimators have been studied extensively in the literature. They have been shown to be minimax optimal density estimators with certain assumptions. But this line of work is not sufficiently discussed in this submission.


## The message is mixed
GANs are implicit generative models that do not have a direct way to access the likelihood information. In this work, the authors considered
both optimizing the latent z and the noisy $p_\lambda$.
The former can be unstable due to the optimization difficulty.
Following Figure 1, the authors stated that "GAN is unable to reconstruct training images when only z is optimized." Is this due to poor optimization and stuck as local minima?
The latter is computationally prohibitive to well-approximate. Motivated by the experiments, the authors identified that the reconstruction term is dominant and have since considered the negative z-reconstruction error.
However, the link between the z-reconstruction error and whether a GAN model captures the density information is not that strong.
For instance, the L2 reconstruction argument is only coming from the added Gaussian noise, which is not inherent to the GAN methodology.



## Analysis is weak
This work is mainly empirical. Although the experiments are interesting, no rigorous argument has been made.
Most of the findings are left on the surface, without digging into the underlying reason. The impact of this work can be greatly improved if the authors can gain a deeper understanding of the training of GAN and propose modifications to it to improve GANs in inference tasks.



Reference:
[1] Liang, Tengyuan. "How well generative adversarial networks learn distributions." The Journal of Machine Learning Research 22.1 (2021): 10366-10406.

[2] Uppal, Ananya, Shashank Singh, and Barnabás Póczos. "Nonparametric density estimation & convergence rates for gans under besov ipm losses." Advances in neural information processing systems 32 (2019).

**Questions:**

See weakness

---

> ### Author Response · Authors · 2023-11-14
> **Response to Reviewer PGsS**
>
> We thank the reviewer for their comments and suggestions.
>
> Regarding the reviewer’s statement on the motivation of our paper, as the reviewer stated, similar results to ours have been shown for different generative models, such as VAEs, normalizing flows and autoregressive models. However, none of these models can generate  images as diverse and realistic as modern GANs do on challenging datasets such as ImageNet. In our mind, this makes it natural to ask whether modern GANs are good density models.
>
> Please see the general comment made to all reviewers for a more detailed discussion regarding the motivation. Briefly, as the reviewer mentioned, GANs are rarely used for tasks such as classification and outlier detection. But GANs _are_ used for other inference tasks, such as image superresolution, medical image segmentation, augmentation for downstream models and more. In all of these applications it is vital to accurately capture the true distribution of the data, or at least to be aware of the fact that the GAN does _not_ capture this distribution.
>
>
> We thank the reviewer for the suggested related works and will incorporate them in future versions.
>
> ---
>
> “In this work, the authors considered both optimizing the latent z and the noisy $p_\\lambda$.”
> Please note that we _do not_ optimize the noisy distribution $p_\\gamma(x)$. Given a GAN generator, a value for $\\gamma$ is chosen and fixed and defines a new distribution. Moreover, the $p_\\gamma(x)$ distribution plays no role during the z-reconstruction. During z-reconstruction, we use gradient descent with multiple random restarts on the latent space $z$ in order to optimize $\\|G(z)-x\\|^2$ - $\\gamma$ plays no role in this. We hope this clears up any misunderstanding there might have been.
>
> We disagree with the reviewer’s comment with regards to the strength between z-reconstruction and the likelihood.  In particular, the claim that “the L2 reconstruction argument is only coming from the added Gaussian noise” is incorrect. In fact, in the noiseless case, any image that cannot be directly reconstructed has zero density so that the z-reconstruction error is by far the dominant term in the likelihood function.

---

> > ### Comment · Reviewer_PGsS · 2023-11-22
> >
> > Thanks for the response. Some of my concerns are addressed, but the motivation is still not convincing. I will keep my score on this version.

---

### Official Review · Reviewer_pQQy · 2023-11-06

**Soundness:** 3 good
**Presentation:** 3 good
**Contribution:** 2 fair
**Rating:** 3
**Confidence:** 4

**Summary:**

The paper seeks to provide an explanation for why GANs remain poor density models despite having impressive sampling capabilities. The authors use Annealed Importance Sampling (AIS) to estimate log likelihoods for GANs. In their experiments on pretrained models such as StyleGAN and BigGAN, they find that these models assign high likelihood scores to seemingly out of distributions samples, implying that outlier detection via GANs might be hard. Additionally, they also show that images reconstructed by searching through z-space to be similar to training examples are given low likelihood scores, implying that GANs seem to not learn the underlying distribution of the training data.

**Strengths:**

1. The main idea is pretty straightforward and the paper is nicely written with clear explanations. (Some concerns discussed below)
2. Significance: understanding failure of distribution learning in GANs is a significant problem especially in more modern architectures.

**Weaknesses:**

1.  My main concern regarding this paper lies with novelty. That GANs don’t learn the underlying distribution has been known for some time (see Arora et al. [2]) so while it is nice to have confirmation of this fact, it does not add to our understanding of GANs. What would be interesting is a new explanation for WHY these models behave the way they do. The authors do attempt to provide an explanation via log likelihood estimation but I have some concerns about the methodology and results. (See next point)
2. The authors report that the GANs should be, based on their results, bad at outlier detection and that they do not memorize training samples. However, prior works on GANs have reported the exact opposite results in both these cases. There is large body of literature on GANs being used for anomaly detection [3] and there is work demonstrating both theoretically [4] and empirically [5] that GANs tend to memorize training samples. I am thinking this might be because the estimated log likelihood used by the authors is not actually a good estimate of underlying log likelihood of the model itself. I am willing to believe that the authors might have found something that previous works have missed and am open to be corrected on this.
3. Writing wise, while I think most of the paper is clear about the mechanics of the experiments, I would encourage the authors to summarize their contributions clearly at the start of the paper (preferably at the end of the introduction). At several points during reading the paper, I was confused about the exact contributions of the current paper vs contributions from prior work.
4. Related to the writing point above, the paper is titled ‘Why are Modern GANs poor density models?’ but I’m still not sure about the exact reason the authors are proposing to be the cause for this phenomenon (poor density modelling). I would encourage the authors to clearly state the reason. The closest I found was in the abstract: “ To try and understand the source of this poor performance, we show that the likelihood that a GAN assigns to an input image is dominated by the quality of the GAN reconstruction when only the latent variable is optimized ”. This is then again touched on in Section 4 briefly. I would have liked some more analysis about this proposed reason. One such question for example is, do all training samples have similarly bad likelihood? Which modes are represented? (Mode collapse)
5. Dataset size is limited. The authors report log likelihood on on 200 samples from only 10 classes on ImageNet (along with results on small samples from other datasets). Classes used are mentioned din the appendix, however, it is unclear if the results in the paper are merely an artifact of the selection of classes made since there is no justification given for why these classes were chosen. I understand that not everyone has access to large compute resources but the concern remains if there is no ablation/justification.

[1] Wu et al. On the quantitative analysis of decoder-based generative models

[2] Arora et al. Do GANs learn the distribution? Some Theory and Empirics

[3] Mattia et al. A Survey on GANs for Anomaly Detection

[4] Nagarajan et al. Theoretical Insights into Memorization in GANs

[5] Bai et al. Reducing Training Sample Memorization in GANs by Training with Memorization Rejection

**Questions:**

I am intrigued (and a little confused) by the finding that GANs used by the authors don’t seem to memorize training examples seemingly in contradiction of prior work. What do the authors think the reason might be?

---

> ### Author Response · Authors · 2023-11-14
> **Response to Reviewer  pQQy**
>
> We thank the reviewer for their comments and suggestions.
>
> As we understand, the major concerns of the reviewer are the novelty of our work and the difference between our results and that of previous works. Regarding the first point, please see the general comment we made to all reviewers regarding what we perceive as the significance of our study.
>
> The second point can be separated into GANs as outlier detectors and memorization in GANs:
> - We used outlier detection as an inference task to analyze the performance of GANs as density estimators. While the reference given by the reviewer is indeed relevant (and will be added to future versions), all methods for anomaly detection with GANs utilize further components that are not part of the generative model of the GAN. On the other hand, if the GAN were a good density estimator, the generator alone could be utilized for anomaly detection (through likelihood calculations).
> - We thank the reviewer for the references regarding memorization and will add them to the related works in future versions. Please note that the authors of reference [4] showed that 7 2-dimensional data points were memorized by a GAN using _tens of thousands of latent codes_. On the other hand, the GANs we analyze model many high-dimensional images, which were augmented during training, so the number of latent codes observed during training (per data point) was much smaller than in [4]. On the other hand, reference [5] analyzes the distribution of nearest neighbors of generated and test samples, showing that the generated samples are slightly closer to the training distribution than the test samples. This could be true even when the GAN can’t _exactly_ reproduce training samples but only generates images that are somewhat more similar to the training data than the test, for instance when there is mode collapse. Conversely, reproducing a training example exactly is a much stronger criterion than being more similar. Does that resolve your concerns regarding memorization?
>
> Answering the reviewers other questions:
>
> 3. We will take the reviewer’s suggestion and will indeed add a contribution list in future versions. Thank you!
>
> 4. Our view is that GANs are bad density estimator because they assign higher likelihood to images outside the data distribution than to training images, in particular SVHN/flat images. Moreover, the conditional versions of the GANs typically assign the same likelihood to all classes - for instance, StyleGAN2-ADA trained on AFHQCats assigns the same likelihood to images of cats as to images of dogs (this is the reason it’s classification accuracy is close to 50% in figure 3). To try and explain why this happens, we show that the likelihood is largely affected by the GAN’s z-reconstruct quality. The surprising (and in our opinion, interesting) phenomenon is that GANs reconstruct flat/SVHN images much better than training images, which is counter to the intuition that GANs learn the correct data manifold. Moreover, the reconstruction quality of conditional GANs isn’t greatly affected by which class is conditioned on - whether it is truly the correct class or not (like in the cats vs dogs example given earlier). This explains why all GANs had bad performance on OD and generative classification. Regarding what, during training, causes the GAN to learn these solutions - that is a very interesting question but well outside the scope of our submission. Moreover, as the reviewer suggested, this method could be used in order to analyze more fine-grained statistics in GANs, for instance whether some training examples have higher likelihood than others, but this (again) is outside the scope of this submission. Does this answer your questions?
>
> 5. We wholeheartedly agree with the reviewer that an analysis on further data would be optimal. In this case, we believe that the extremely poor performance of GANs on classification and outlier detection do give a good indication of their problems. The classes used in ImageNet10 were chosen ahead of time, before any experiments were conducted on ImageNet images.
>
>
> [4] Nagarajan et al. Theoretical Insights into Memorization in GANs
>
> [5] Bai et al. Reducing Training Sample Memorization in GANs by Training with Memorization Rejection

---

> > ### Comment · Reviewer_pQQy · 2023-11-23
> > **Response to rebuttal**
> >
> > I thank the authors for their response. However, my concerns regarding novelty (also raised by other reviewers) remain and so I will maintain my score.

---

### Official Review · Reviewer_GHc8 · 2023-11-06

**Soundness:** 2 fair
**Presentation:** 2 fair
**Contribution:** 2 fair
**Rating:** 3
**Confidence:** 3

**Summary:**

The authors present a computational study that evaluates GANs as density models. They first introduce a "relaxed" likelihood which they propose to use for likelihood estimation using annealed importance sampling (AIS). They then use the AIS likelihood estimates to compute, among other things, GAN classification accuracy or out-of-distribution detection performance.

**Strengths:**

- The paper is an interesting study on the capability of GANs as density models.
- The paper is well written and generally easy to follow.
- The paper fits well in line with the recent literature on "out-of-distribution detection" of neural density models and should be a valuable contribution to the field.

**Weaknesses:**

Overall, the paper has several limitations which are discussed below:

-  The paper seems to be of only limited originality and novelty. It builds on the work by Wu et al. [1] to use AIS to compute likelihood estimates for generative models, and uses it to evaluate the capability of GANs as density estimators in several experiments.

- Conceptually, some aspects of the manuscript are unclear to the reviewer: what is the motivation to frame GANs as density models, since vanilla GANs are not trained with, e.g., maximum (marginal) likelihood objectives. Why would the adversarial loss of GANs lead to good density models and what did the authors expect to find with this study? Can the poor performance of GANs for density estimation tasks be remedied? How does the authors' work fit in the context of related recent literature, such as [2-4].

- As an empirical computational study, the evaluations are a bit limited, and the paper is sometimes difficult to follow and/or lacks technical details and clarity. For instance:
    - The influence of the stochastic relaxation $p(x) = \int p(z) p(x | G(z))dz$ on the density estimation is not sufficiently discussed. Could the introduction of the noise lead to lower (or vice versa higher) likelihoods and not the model itself? How is the noise variance chosen?
  - In Section 2.2, the authors write "[i]n generative classification, it is assumed that different parametric distributions
were learned for each class" and define a conditional likelihood $p_\theta(x|c)$. How does this likelihood look like? Do the authors mean something like $p_{\theta_c}(x)$ where for each class a density model is learned, or how is the conditioning done?
  - Figure 4 shows that the style GAN estimates higher likelihoods to flat and SVHN data than to training data. It is not clear if this result is just an artifact, because the intrinsic dimension of flat/SVHN data is significantly lower than the intrinsic dimension of the training/testing data (CIFAR10). What happens when applied to a data set with higher intrinsic dimension like ImageNet?
  - Equation 6 introduces a conditional density $p(z|x)$.  It is not clear how this distribution is computed. Also, the equality does apparently not hold. As far as I can tell, the right-hand-side is the evidence lower bound, and hence it is an inequality.

- Given that the paper is entirely empirical, the code base is not of sufficient quality: which libraries are used and which versions do they have, how can the results be reproduced, etc.

[1] https://arxiv.org/abs/1611.04273
[2] https://proceedings.neurips.cc/paper_files/paper/2018/hash/4996dcc43b5be197b5887a4e60817b1c-Abstract.html
[3] https://proceedings.neurips.cc/paper/2019/hash/959ab9a0695c467e7caf75431a872e5c-Abstract.html
[4] https://openaccess.thecvf.com/content_CVPR_2019/html/Abbasnejad_A_Generative_Adversarial_Density_Estimator_CVPR_2019_paper.html

**Questions:**

Questions:

- How is the variance of $\eta$ in Equation 1 chosen? This seems to be a critical hyperparameter.
- How did you select the classes in ImageNet10 and CIFAR10?
- How are the baselines fitted? Have KDEs/diagonal Gaussians been fitted for each class or have they been fitted on the pooled data?

---

> ### Author Response · Authors · 2023-11-13
> **Response to Reviewer GHc8**
>
> We thank the reviewer for their comments and suggestions.
>
> From what we understand, the major concerns of the reviewer were:
> 1. It is not clear whether GANs should even learn a good density model at all.
> 2. The significance and novelty of the study with regards to the research of Wu et al.
>
> For these concerns, please see the general comment made regarding GANs as density models and the significance of our research. Briefly, it has been shown that GANs should theoretically learn optimal density models and we hoped that this would be the case - it isn’t, even for state-of-the-art GANs with low FID scores, high sample quality and little to no mode collapse.
> In particular, we are not attempting to suggest a new density model, only to analyze existing GANs.
>
>
> ---
>
> Answering the reviewer's more specific questions:
> - The reviewer is correct, the variance of the stochastic relaxation can indeed influence the relaxed likelihood. As written in section 3, we chose a value for the variance that maximizes this likelihood, which is equal to the variance of the z-reconstruction error.
> - As the reviewer mentioned, generative classification assumes a different parametric model is trained for each class. When using conditional GANs, we used the conditioning of conditional GANs.
> - Equation 6 is an equivalence, but because it seems that it is not well known, we will add a derivation to the appendix. To see that equation 6 is correct (as long as $p(x)>0$), note that:
> $$
>     D_{\\text{KL}}\\left(p(z|x)||p\\left(z\\right)\\right)  =\\intop p(z|x)\\log\\frac{p(z|x)}{p\\left(z\\right)}dz
>   =\\intop p(z|x)\\left[\\log\\frac{p(z|x)}{p\\left(z\\right)}+\\log p(x)-\\log p(x)\\right]dz
>   =\\mathbb{E}_{z|x}\\left[\\log p(x|z)\\right]-\\log p(x)
> $$
> - The classes for ImageNet10 are a subset of 10 classes from the full ImageNet, which were chosen before any experiments were performed. In CIFAR10, all classes were used.
> - As the reviewer mentioned, the baselines were trained on each class separately.

---

> > ### Comment · Reviewer_GHc8 · 2023-11-20
> >
> > I thank the authors for clarifications and responding to my questions. As the authors point out, understanding the capabilities of GANs as density estimator is certainly critical for several applications.
> >
> > However, the author's clarifications still don't warrant a higher rating which is why I will retain my score.

---

### Author Response · Authors · 2023-11-13
**Comment for all reviewers**

We thank the reviewers for their comments and suggestions.


A major point made in the reviews is that GANs should not be expected to be good density models of the training data because they were not trained to be. While we agree that most GANs were not specifically designed for density estimation, it has been shown theoretically that optimally trained GANs learn the true distribution $p_\text{data}(x)$ (e.g. [1]). This means that, in theory, $p_\theta(x) = p_\text{data}(x)$ where $\theta$ are the parameters of the generator. From these statements, theoretically GANs should be optimal density estimators. Whether it is reasonable to expect this behavior or not in practical applications, the goal of all the papers introducing new GANs is to model the density of the training distribution. As such, we believe checking whether this truly is the case is interesting.


Another question raised is whether it is important that GANs are good density estimators or not, as they are not primarily used for inference tasks such as classification and outlier detection. It is important to note that GANs are used in broader contexts than only image generation. For instance, they are used to aid segmentation in medical applications [4] or as a method to generate more data for training other models [5,6], among many others. In all such applications, whether the GAN correctly learns the training distribution or not is of vital importance.


Furthermore, as the reviewers mentioned, past research showed that certain GANs did not learn the training distribution [2, 3]. This was due to low sample quality, mode collapse or duplication of training examples. However, the mentioned research is from _more than 5 years ago_ and GANs have drastically improved over these years. In particular, modern GANs are able to generate remarkably realistic samples for complex datasets such as ImageNet, they no longer suffer from “mode collapse” (evidenced by their very low FID scores) and they no longer copy training examples (see e.g. figure 10 of [7]).


These improvements seem to suggest that state-of-the-art GANs might actually capture the data distribution. We believe that finding out whether they do or do not is of significance. In our submission we show definitive indications that, even with the new improvements, state-of-the-art GANs do not learn the correct distribution.


As the above discussion is clearly missing from the main text, we will add it to the introduction in future versions.


[1]: Goodfellow, Ian, et al. "Generative adversarial nets." Advances in neural information processing systems 27 (2014).

[2]: Wu, Yuhuai, et al. "On the quantitative analysis of decoder-based generative models." (2016).

[3]: Arora, Sanjeev, Andrej Risteski, and Yi Zhang. "Do GANs learn the distribution? some theory and empirics." International Conference on Learning Representations. 2018.

[4]: Skandarani, Youssef, Pierre-Marc Jodoin, and Alain Lalande. "Gans for medical image synthesis: An empirical study." Journal of Imaging 9.3 (2023): 69.

[5]: Wang, Yu-Xiong, et al. "Low-shot learning from imaginary data." Proceedings of the IEEE conference on computer vision and pattern recognition. 2018.

[6]: Figueira, Alvaro, and Bruno Vaz. "Survey on synthetic data generation, evaluation methods and GANs." Mathematics 10.15 (2022): 2733.

[7]: Karras, Tero, et al. "Analyzing and improving the image quality of stylegan." Proceedings of the IEEE/CVF conference on computer vision and pattern recognition. 2020.

---

### Meta-Review · Area_Chair_vhNR · 2023-12-10

**Metareview:**

In this paper, the authors focus on answering why GANs are poor density models. But this fact is already well-known, even when the motivation for GANs to be the universal simulator. This result could be achieved by matching the relevant statistics and not fully modeling the distribution. The paper takes from the existing literature, as has been pointed out by the reviewers, on the limitations of GANs (and other generative models). However, the paper does not present new insights on why they aren't good enough for density estimation or on how to fix it. The authors need to ensure that they provide new insights or be able to publish this paper at a major conference.

**Justification For Why Not Higher Score:**

The paper has nothing new or insightful.

**Justification For Why Not Lower Score:**

not applicable

---

### Decision · Program_Chairs · 2024-01-16

Reject